

# A functional-analysis derivation of the parquet equation

Christian J. Eckhardt[1,2,3]⋆, Patrick Kappl[2], Anna Kauch[2] and Karsten Held[2]

**1** Institute for Theoretical Solid State Physics,
RWTH Aachen University, 52074 Aachen, Germany
**2** Institute of Solid State Physics, TU Wien, 1040 Vienna, Austria
**3** Max Planck Institute for the Structure and Dynamics of Matter, 22761 Hamburg, Germany

⋆ christian.eckhardt@mpsd.mpg.de

## Abstract

The parquet equation is an exact field-theoretic equation known since the 60s that underlies numerous approximations to solve strongly correlated Fermion systems. Its derivation previously relied on combinatorial arguments classifying all diagrams of the two-particle Green's function in terms of their (ir)reducibility properties. In this work we provide a derivation of the parquet equation solely employing techniques of functional analysis namely functional Legendre transformations and functional derivatives. The advantage of a derivation in terms of a straightforward calculation is twofold: (i) the quantities appearing in the calculation have a clear mathematical definition and interpretation as derivatives of the Luttinger–Ward functional; (ii) analogous calculations to the ones that lead to the parquet equation may be performed for higher-order Green's functions potentially leading to a classification of these in terms of their (ir)reducible components.


## 1  Introduction

The classification of diagrams in terms of their two-particle (ir)reducibility started with the seminal work by Salpeter and Bethe in 1951 [1], published just two years after Feynman's [2,3] invention of the diagrammatic technique and Dyson's [4,5] related concepts of one-particle (ir)reducibility. A decade later, in 1964, De Dominicis and Martin [6,7] classified the two-particle diagrams further into irreducible and three distinct types of reducible diagrams: the famous parquet equation. Here, De Dominicis and Martin used combinatorial arguments and also a Legendre transform of the free energy in order to analyze irreducibility properties.

The parquet equation supplemented by Bethe–Salpeter equations [1] in the three channels to generate the reducible diagrams, and the Dyson [4,5] plus Schwinger-Dyson equation [8] to connect two- and one-particle diagrams constitute an exact set of equations. It allows one to calculate all one- and two-particle functions if the two-particle fully irreducible vertex is known (or approximated). On the downside doing such parquet calculations is rather involved, and thus parquet calculations have been rarely done in the decades following their invention. An outstanding exception is the first order parquet calculation that solves the x-ray problem [9].

With the advent of computers, there has been a renewed push for a now numerical solution of the parquet formalism in the early 1990s by Bickers *et al.* [10–13]. One motivation for using the parquet equation to treat systems of strongly correlated electrons stems from arguments that the so-called *parquet approximation*, where the fully irreducible 2-particle vertex is approximated by the bare interaction, fulfills the Mermin–Wagner theorem [11]. First numerical evidence that this is indeed the case was presented in Ref. [14]. However, efforts to numerically employ the parquet approximation were eventually abandoned because the involved equations remained numerically too demanding. This changed more recently as, thanks to Moore's law, still limited but already meaningful parquet calculations became possible [14–21]. As a further improvement of the method, using all local diagrams for the fully irreducible vertex [14, 16, 22] instead of the bare Coulomb interaction as in the previously employed *parquet approximation* represents a better, non-perturbative starting point.

In the present paper, we will re-derive the parquet equation using the free energy functional and Legendre transformation instead of the original combinatorial arguments. This more systematic derivation may make it easier to transfer the concept of the parquet equation from the two- to the three-particle level, a research field which has gained some interest recently [23–25]. Using functional derivatives and Legendre transforms is a common approach to derive equations and develop new methods in quantum field theory, see, e.g., [26–30]. However, to the best of our knowledge, it has not been employed hitherto to derive the parquet equation.

The manuscript is organized as follows: In Section 2 we introduce $\Phi^4$ theory for which we define the partition function and the free energy. We recap how the connected Green's functions can be obtained from the free energy via functional derivatives. In order to make this work self-contained, we compute the first Legendre transformation of the free energy, also called vertex generating function, in Section 3 roughly following the steps of Ref. [30] Chs. 11 and 12. We outline how a particular property of the Legendre transformation can be used in order to relate Green's functions and vertex functions which then yields Dyson's equation. Finally, Section 4 contains the main result of the paper. We compute the second

Legendre transformation of the free energy, known as the Luttinger–Ward functional, and show that using the same property of the Legendre transformation that previously yielded Dyson's equation can in the case of the second transformation be used to derive the parquet equation.

## 2 Connected diagrams

We consider $\Phi^4$ theory that is defined through the action

$$S[\Phi] = -\sum_{n=1}^{4} \frac{1}{n!} V_n^{a_1 \ldots a_n} \Phi^{a_1} \ldots \Phi^{a_n}, \tag{1}$$

using Einstein's summation convention here and throughout the paper. $\Phi$ denotes fields that depend on some set of quantum numbers $a_1 \ldots a_n$ which can be discrete, continuous, or a mixture of both. $V_n$, $n = 1, 2, 4$ denote the bare potentials defining the theory. We assume these to be symmetric in all their arguments. In particular $V_1$ constitutes a source term while $V_2 = -G_0^{-1}$ is the inverse bare Green's function. $V_4$ is a scattering term. The partition function of a field theory is defined as

$$Z = N \int \mathrm{D}\Phi e^{-S}, \tag{2}$$

where the constant $N$ is chosen for correct normalization and $\int \mathrm{D}\Phi$ denotes the Feynman path integral. The partition function $Z$ can be represented by summing all connected and disconnected Feynman diagrams [29, 30]. A central goal to learn something about the field theory at hand is to compute expectation values. For this it is convenient to take the logarithm of the partition function

$$W := \ln Z. \tag{3}$$

$W$ is commonly called the free-energy functional. It depends on the bare potentials, $V_n$ with $n = 1, \ldots, 4$, and can be represented by connected Feynman diagrams and a trivial term [7, 29]:

$$W = \frac{1}{2} \ln\left(-V_2^{-1}\right) + \ldots \text{ connected diagrams} \ldots \tag{4}$$

The fact, that only connected Feynman diagrams need to be considered for the computation of $W$ constitutes a substantial simplification since the connected diagrams for $W$ are a subset of all diagrams, connected and disconnected, that need to be considered for the computation of the partition function $Z$. The diagrams for $W$ up to second order in the scattering potentials for the here considered $\Phi^4$ theory are shown in Fig. 1. Expectation values of the field operators $\langle \Phi^n \rangle$, that constitute observables that one would usually like to compute, can be calculated by performing functional derivatives of the free-energy functional with respect to the bare potentials $V_n$

$$\frac{1}{n!} \langle \Phi^n \rangle = \frac{\delta W}{\delta V_n} =: \widetilde{G}_n, \tag{5}$$

where we have introduced $\widetilde{G}_n$ as the $n$-point expectation value or equivalently the $n$-point (connected and) disconnected Green's function up to a prefactor in order to be consistent with literature [30]. On the other hand, by repeatedly performing derivatives of W with respect to $V_1$, one obtains the connected $n$-point ($n/2$-particle) Green's functions $G_n$ [30, 31]

$$\frac{\delta^n W}{(\delta V_1)^n} =: G_n, \tag{6}$$

$$W - \frac{1}{2}\ln\left(-V_2^{-1}\right) = \frac{1}{2} + \frac{1}{24} + \frac{1}{4} + \frac{1}{8} + \frac{1}{16}$$

$$+ \frac{1}{16} + \frac{1}{8} + \frac{1}{8} + \frac{1}{12}$$

$$+ \frac{1}{48} + \frac{1}{16} + \frac{1}{72} + \mathcal{O}(V_4^3)$$

Figure 1: $W$ represented by diagrams in $\Phi^4$ theory up to second order in $V_4$. Here the lines denote bare propagators. The diagrammatic rules used in this paper are summarized in Appendix D.

where $\frac{\delta^n}{(\delta V_1)^n}$ denotes the $n^{\text{th}}$ functional derivative with respect to $V_1$. The reason these Green's functions are connected is that each functional derivative $\frac{\delta}{\delta V_1}$ removes one external one-point vertex in the diagrams for $W$ (see Fig. 1) leaving the initial connectedness of $W$ intact. These connected Green's functions can be identified with the cumulants of the field theory [30], rather than the moments as in the case of $\widetilde{G}_n$. For example, we have $G_2 = \langle \Phi^2 \rangle - \langle \Phi \rangle^2$. Computing $G_n$ for all $n > 0$ would correspond to a cumulant expansion of the respective theory [30] and thus constitute the full information about the theory. Furthermore, the disconnected Green's functions $\widetilde{G}_n$, i.e., the expectation values of field operators, can be calculated from the connected ones $G_n$, by simply adding all disconnected parts which is usually straightforward. Therefore, it is the general goal of this work to identify relations that enable the computation of the $G_n$ or find accurate approximations for it.

At this point we also mention that throughout the paper we use the interchangeability of functional derivatives as, e.g., in

$$\frac{\delta}{\delta V_1^a}\left(\frac{\delta W}{\delta V_1^b}\right) = \frac{\delta}{\delta V_1^b}\left(\frac{\delta W}{\delta V_1^a}\right), \tag{7}$$

and similarly for all other occurring functional derivatives. This holds for the here considered functionals due to their polynomial form.

## 3 One-particle irreducible diagrams

We perform a variable substitution in the free-energy functional $W$ according to

$$V_1 \rightarrow \frac{\delta W}{\delta V_1}. \tag{8}$$

For this substitution the Legendre transformation is the most common tool. We thus define the first Legendre transform of the free-energy functional as

$$\Lambda_1 := W - V_1^a \frac{\delta W}{\delta V_1^a}. \tag{9}$$

From now on we also abbreviate the quantum number indices $a_1, a_2, a_3$, etc. with $a$, $b$, $c$, etc. The newly introduced $\Lambda_1$ is a functional of $\widetilde{G}_1 (= G_1)$, $V_2$, and $V_4$. It is possible to show that $\Lambda_1$ can be represented by one-particle irreducible (1PI) diagrams (see Fig. 2) plus a trivial term as shown in [7,30,32]:

$$\Lambda_1 - \frac{1}{2}\ln\left(-V_2^{-1}\right) = \frac{1}{2}\ \ \text{+}\ \frac{1}{24}\ \ \text{+}\ \frac{1}{4}\ \ \text{+}\ \frac{1}{8}\ \ \text{+}\ \frac{1}{16}$$

$$+ \frac{1}{8}\ \ \text{+}\ \frac{1}{12}\ \ \text{+}\ \frac{1}{48}\ \ \text{+}\ \frac{1}{16}\ \ \text{+}\ \mathcal{O}(V_4^3)$$

Figure 2: $\Lambda_1$ represented by diagrams in $\Phi^4$ theory up to second order in $V_4$. Lines connecting interaction vertices denote the bare propagator. Other diagrammatic rules are summarized in Appendix D.

$$\Lambda_1 = \frac{1}{2}\ln\left(-V_2^{-1}\right) + \dots \text{1PI diagrams} \dots \tag{10}$$

This might simplify computations since the diagrams for $\Lambda_1$ are a real subset of the diagrams for $W$ namely only the one-particle irreducible ones. 1PI $n$-point functions can be computed by performing the $n^{\text{th}}$ functional derivative of $\Lambda_1$ with respect to $\widetilde{G}_1$:

$$\frac{\delta^n \Lambda_1}{(\delta\widetilde{G}_1)^n} =: C_n. \tag{11}$$

If it is possible to relate the 1PI $n$-point functions $C_n$ to the connected Green's functions $G_n$, one can potentially perform computations with the simpler object $\Lambda_1$ rather than $W$. This can indeed be done using basic properties of the Legendre transformation. The first one reads [29,30]

$$\frac{\delta\Lambda_1}{\delta\widetilde{G}_1} = -V_1. \tag{12}$$

The second equation is an inverse-second-derivative relation which can be derived as [29,30]

$$\delta^{ab} = \frac{\delta V_1^b}{\delta V_1^a} = -\frac{\delta}{\delta V_1^a}\frac{\delta\Lambda_1}{\delta\widetilde{G}_1^b} = -\frac{\delta\widetilde{G}_1^c}{\delta V_1^a}\frac{\delta^2\Lambda_1}{\delta\widetilde{G}_1^c\,\delta\widetilde{G}_1^b} = -\frac{\delta^2 W}{\delta V_1^a\,\delta V_1^c}\frac{\delta^2\Lambda_1}{\delta\widetilde{G}_1^c\,\delta\widetilde{G}_1^b} = -G_2^{ac}C_2^{cb}, \tag{13}$$

where we employed the functional chain rule as well as Eqs. (5), (6), (11) and (12). Rearranging yields

$$G_2^{ab} = -(C_2^{-1})^{ab}. \tag{14}$$

We can identify Eq. (14) as the well-known Dyson equation by noting that $C_2$ can be split up in two parts: one containing no interaction vertices originating from the derivative of the first term in Eq. (9), and all other terms each containing at least one interaction term:

$$C_2 = -\left(G_0\right)^{-1} + C_2^{\text{int}}. \tag{15}$$

Inserting this into Eq. (14) yields [5,31]

$$G_2^{ab} = \left(\left(\left(G_0\right)^{-1} - C_2^{\text{int}}\right)^{-1}\right)^{ab}, \tag{16}$$

which is the Dyson equation in its more familiar form, and we can identify $C_2^{\text{int}} =: \Sigma$ as the self-energy $\Sigma$ of the theory. This derivation of Dyson's equation is textbook knowledge – see, e.g., Ref. 30 Eq. (12.7). It is consistent with the notion that the self-energy of a theory consists of all 1PI two-point functions including at least one interaction term [26,30].

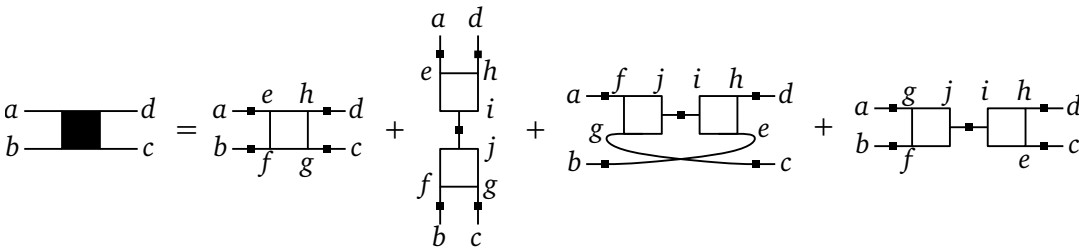

Figure 3: Feynman diagrams for Eq. (17). The diagrammatic rules used in this paper are summarized in Appendix D.

Figure 4: Graphic illustration of Eq. (18) via Feynman diagrams. This already resembles the parquet equation, but classifies diagrams in terms of one-particle reducibility instead of two-particle reducibility. In the here considered $\phi^4$-theory the appearing one-particle reducible terms will be zero by symmetry. The diagrammatic rules used in this paper are summarized in Appendix D.

To obtain higher-order connected Green's functions in terms of 1PI functions, one can perform derivatives of Eq. (14) with respect to $V_1$. This is, e.g., outlined in Ref. 30 Ch. 12. Again, using the functional chain rule as well as Eqs. (6) and (11) yields

$$\underbrace{\frac{\delta}{\delta V_1^c} G_2^{ab}}_{=G_3^{abc}} = -\frac{\delta}{\delta V_1^c}(C_2^{-1})^{ab} = -(C_2^{-1})^{ae}\frac{\delta}{\delta V_1^c}C_2^{ef}(C_2^{-1})^{fb} = G_2^{ae}G_2^{cg}C_3^{efg}G_2^{fb}. \tag{17}$$

The result is shown diagrammatically in Fig. 3. We see that in order to compute the connected three-point Green's function $G_3$ from the 1PI three-point function $C_3$ one needs to attach a connected two-point Green's function to each external leg of $C_3$.

To get the connected four-point Green's function $G_4$, we perform another derivative on Eq. (17):

$$\overbrace{\frac{\delta}{\delta V_1^d} G_3^{abc}}^{=G_4^{abcd}} = \frac{\delta}{\delta V_1^d}\left(G_2^{ae}G_2^{cg}C_3^{efg}G_2^{fb}\right) \tag{18}$$

$$= G_2^{ae}G_2^{cg}C_4^{efgh}G_2^{fb}G_2^{hd} + G_2^{ae}G_2^{dh}C_3^{ehi}G_2^{ij}C_3^{jfg}G_2^{fb}G_2^{gc}$$

$$+ G_2^{be}G_2^{dh}C_3^{ehi}G_2^{ij}C_3^{jfg}G_2^{fa}G_2^{gc} + G_2^{ce}G_2^{dh}C_3^{ehi}G_2^{ij}C_3^{jfg}G_2^{fb}G_2^{ga}.$$

This equation is depicted diagrammatically in Fig. 4. Equation (18) shows that the connected four-point Green's function can be obtained by attaching connected two-point Green's functions to the four external legs of the 1PI four-point function $C_4$ and adding three different diagrams which are one-particle reducible. Here, one-particle reducibility refers to diagrams that can be cut into two parts by cutting a single Green's function line.

In the here considered $\phi^4$ theory, explicitly excluding a $\phi^3$ term, we have $G^3 = 0$ and $C^3 = 0$. However, in order to derive Eq. (18) one must only set such quantities to zero at the end of the calculation, after having performed all possible derivatives. Doing this will cancel the one-particle reducible terms in Eq. (18).

$$\Lambda_2^{\text{int}} = \frac{1}{2}\; \text{∎∎} \; + \frac{1}{24}\; \text{⧓} \; + \frac{1}{4}\; \text{∎⌀∎} \; + \frac{1}{8}\; \text{∞} \; + \frac{1}{12}\; \text{∎⬡∎} \; + \frac{1}{48}\; \text{⬭} \; + \mathcal{O}(V_4^3)$$

Figure 5: Diagrammatic representation of $\Lambda_2^{\text{int}}$ up to second order in $V_4$. Here, lines connecting interaction vertices denote the full propagator $G_2$ for which we have not included an explicit extra vertex for brevity. Other diagrammatic rules are summarized in Appendix D.

## 4 Two-particle irreducible diagrams and parquet equation

We perform the second Legendre transformation of the free energy [29]

$$\Lambda_2 := W - V_1^a \frac{\delta W}{\delta V_1^a} - V_2^{ab} \frac{\delta W}{\delta V_2^{ab}}\,. \tag{19}$$

$\Lambda_2$ is a functional of $\widetilde{G}_1, \widetilde{G}_2$ and $V_4$ and is given by all two-particle irreducible (2PI) diagrams plus a trivial term [7,29,32]

$$\Lambda_2 = \frac{1}{2}\ln G_2 + \dots \text{2PI Diagrams} \dots \tag{20}$$

Note that the trivial term now contains the connected two-point function instead of the bare Green's function as has been shown in [7]. We define the part of $\Lambda_2$ as the one containing at least one interaction vertex

$$\Lambda_2^{\text{int}} := \Lambda_2 - \frac{1}{2}\ln G_2\,. \tag{21}$$

$\Lambda_2^{\text{int}}$ is the Luttinger–Ward functional of the considered $\Phi^4$ field theory [29]. Figure 5 shows its diagrammatic representation up to second order in $V_4$. Again, we use properties of the Legendre transformation in order to relate 2PI quantities back to the connected Green's functions. The first two such relations, analogous to Eq. (12), read [29,30]

$$\frac{\delta \Lambda_2}{\delta \widetilde{G}_1} = -V_1\,, \tag{22}$$

and

$$\frac{\delta \Lambda_2}{\delta \widetilde{G}_2} = -V_2\,. \tag{23}$$

Furthermore, one has a relation of inverse second derivatives analogous to Eq. (13). Since we have replaced two arguments, a derivative relation in both arguments can be written as a $2 \times 2$ matrix containing four elements overall (Hessian matrix) leading overall to four relations that are analogous to Eq. (13). As proposed in Ref. 33, we also introduce the variable substitution $\widetilde{G}_2 \to G_2$ and write $\Lambda_2$ as a function of $G_1 = \widetilde{G}_1$ and $G_2$:

$$\Lambda_2[\widetilde{G}_1, \widetilde{G}_2] = \Lambda_2[\widetilde{G}_1[G_1, G_2], \widetilde{G}_2[G_1, G_2]]\,. \tag{24}$$

This is convenient as one otherwise performs calculations with a disconnected propagator $\widetilde{G}_2 = \frac{1}{2}(G_2 + G_1 G_1)$ which is possible but leads to unnecessary complications.

We can now derive the first inverse-second-derivative relation similarly to Eq. (13):

$$\delta^{ac} = \frac{\delta V_1^c}{\delta V_1^a} = -\left( \frac{\delta G_1^e}{\delta V_1^a} \frac{\delta}{\delta G_1^e} + \frac{\delta G_2^{ef}}{\delta V_1^a} \frac{\delta}{\delta G_2^{ef}} \right) \frac{\delta \Lambda_2}{\delta \widetilde{G}_1^c} = -G_2^{ae} \left( \frac{\delta^2 \Lambda_2}{\delta G_1^e \delta G_1^c} - 2\frac{\delta \Lambda_2}{\delta G_2^{ec}} \right) - G_3^{aef} \frac{\delta^2 \Lambda_2}{\delta G_2^{ef} \delta G_1^c}\,. \tag{25}$$

Here we have employed the chain rule to replace the derivative with respect to the disconnected Green's function by that with respect to the connected ones and used Eq. (B.2) to cancel two terms.

After contracting Eq. (25) with $\left[\left(\frac{\delta^2 \Lambda_2}{\delta G_1^\bullet \delta G_1^\bullet} - 2\frac{\delta \Lambda_2}{\delta G_2^{\bullet\bullet}}\right)^{-1}\right]^{cb}$ from the right and reordering terms we get

$$G_2^{ab} = -\left[\left(\frac{\delta^2 \Lambda_2}{\delta G_1^\bullet \delta G_1^\bullet} - 2\frac{\delta \Lambda_2}{\delta G_2^{\bullet\bullet}}\right)^{-1}\right]^{ab} - G_3^{aef}\frac{\delta^2 \Lambda_2}{\delta G_2^{ef}\delta G_1^c}\left[\left(\frac{\delta^2 \Lambda_2}{\delta G_1^\bullet \delta G_1^\bullet} - 2\frac{\delta \Lambda_2}{\delta G_2^{\bullet\bullet}}\right)^{-1}\right]^{cb}. \quad (26)$$

Here $\left[\left(\frac{\delta^2 \Lambda_2}{\delta G_1^\bullet \delta G_1^\bullet} - 2\frac{\delta \Lambda_2}{\delta G_2^{\bullet\bullet}}\right)^{-1}\right]^{ab}$ denotes the $(ab)$ component of the quantity obtained by computing the second functional derivative of $\Lambda_2$ with respect to $G_1$, subtracting twice the derivative of $\Lambda_2$ with respect to $G_2$, and then inverting the resulting $2 \times 2$ matrix. By writing the final arguments outside the square bracket and indicating the arguments of the one-point functions by dots, we highlight that

$$\left(\frac{\delta^2 \Lambda_2}{\delta G_1^a \delta G_1^b} - 2\frac{\delta \Lambda_2}{\delta G_2^{ab}}\right)^{-1} \neq \left[\left(\frac{\delta^2 \Lambda_2}{\delta G_1^\bullet \delta G_1^\bullet} - 2\frac{\delta \Lambda_2}{\delta G_2^{\bullet\bullet}}\right)^{-1}\right]^{ab}. \quad (27)$$

Similarly to Eq. (14), Eq. (26) provides a relation between the connected Green's function $G_2$ and derivatives of the second Legendre transform (Luttinger–Ward functional). Since we are ultimately looking for the parquet equation we, again, perform two more derivatives of Eq. (26) with respect to $V_1$, obtaining the connected four-point Green's function $G_4$ on the left-hand side.

The first derivative yields

$$\frac{\delta}{\delta V_1^b}G_2^{ac} = \left[\left(\frac{\delta^2 \Lambda_2}{\delta G_1^\bullet \delta G_1^\bullet} - 2\frac{\delta \Lambda_2}{\delta G_2^{\bullet\bullet}}\right)^{-1}\right]^{ae}\left(\frac{\delta}{\delta V_1^b}\frac{\delta^2 \Lambda_2}{\delta G_1^e \delta G_1^f}\right)\left[\left(\frac{\delta^2 \Lambda_2}{\delta G_1^\bullet \delta G_1^\bullet} - 2\frac{\delta \Lambda_2}{\delta G_2^{\bullet\bullet}}\right)^{-1}\right]^{fc}$$

$$-G_4^{abef}\frac{\delta^2 \Lambda_2}{\delta G_2^{ef}\delta G_1^g}\left[\left(\frac{\delta^2 \Lambda_2}{\delta G_1^\bullet \delta G_1^\bullet} - 2\frac{\delta \Lambda_2}{\delta G_2^{\bullet\bullet}}\right)^{-1}\right]^{gc}$$

$$-G_3^{aef}\left(\frac{\delta}{\delta V_1^b}\frac{\delta^2 \Lambda_2}{\delta G_2^{ef}\delta G_1^g}\right)\left[\left(\frac{\delta^2 \Lambda_2}{\delta G_1^\bullet \delta G_1^\bullet} - 2\frac{\delta \Lambda_2}{\delta G_2^{\bullet\bullet}}\right)^{-1}\right]^{gc}$$

$$-G_3^{aef}\frac{\delta^2 \Lambda_2}{\delta G_2^{ef}\delta G_1^g}\frac{\delta}{\delta V_1^b}\left[\left(\frac{\delta^2 \Lambda_2}{\delta G_1^\bullet \delta G_1^\bullet} - 2\frac{\delta \Lambda_2}{\delta G_2^{\bullet\bullet}}\right)^{-1}\right]^{gc}$$

$$=\left[\left(\frac{\delta^2 \Lambda_2}{\delta G_1^\bullet \delta G_1^\bullet} - 2\frac{\delta \Lambda_2}{\delta G_2^{\bullet\bullet}}\right)^{-1}\right]^{ae}\left[\left(\frac{\delta G_1^g}{\delta V_1^b}\frac{\delta}{\delta G_1^g} + \frac{\delta G_2^{gh}}{\delta V_1^b}\frac{\delta}{\delta G_2^{gh}}\right)\frac{\delta^2 \Lambda_2}{\delta G_1^f \delta G_1^e}\right]\left[\left(\frac{\delta^2 \Lambda_2}{\delta G_1^\bullet \delta G_1^\bullet} - 2\frac{\delta \Lambda_2}{\delta G_2^{\bullet\bullet}}\right)^{-1}\right]^{fc}$$

$$-G_4^{abef}\frac{\delta^2 \Lambda_2}{\delta G_2^{ef}\delta G_1^g}\left[\left(\frac{\delta^2 \Lambda_2}{\delta G_1^\bullet \delta G_1^\bullet} - 2\frac{\delta \Lambda_2}{\delta G_2^{\bullet\bullet}}\right)^{-1}\right]^{gc}$$

$$-G_3^{aef}\frac{\delta^2 \Lambda_2}{\delta G_2^{ef}\delta G_1^g}\frac{\delta}{\delta V_1^b}\left[\left(\frac{\delta^2 \Lambda_2}{\delta G_1^\bullet \delta G_1^\bullet} - 2\frac{\delta \Lambda_2}{\delta G_2^{\bullet\bullet}}\right)^{-1}\right]^{gc}$$

$$-G_3^{aef}\left[\left(\frac{\delta G_1^h}{\delta V_1^b}\frac{\delta}{\delta G_1^h} + \frac{\delta G_2^{hi}}{\delta V_1^b}\frac{\delta}{\delta G_2^{hi}}\right)\frac{\delta^2 \Lambda_2}{\delta G_2^{ef}\delta G_1^g}\right]\left[\left(\frac{\delta^2 \Lambda_2}{\delta G_1^\bullet \delta G_1^\bullet} - 2\frac{\delta \Lambda_2}{\delta G_2^{\bullet\bullet}}\right)^{-1}\right]^{gc}. \quad (28)$$

When performing the remaining derivative, we only keep terms that have an even number of external legs since terms with an odd number will vanish in the end due to symmetry (since

there is no $V^3$). Under these considerations we get

$$
\begin{aligned}
G_4^{abcd} &= \left[\left(\frac{\delta^2\Lambda_2}{\delta G_1^\bullet \delta G_1^\bullet} - 2\frac{\delta\Lambda_2}{\delta G_2^{\bullet\bullet}}\right)^{-1}\right]^{ae} G_2^{bg} \frac{\delta^4\Lambda_2}{\delta G_1^e \delta G_1^f \delta G_1^g \delta G_1^h} \left[\left(\frac{\delta^2\Lambda_2}{\delta G_1^\bullet \delta G_1^\bullet} - 2\frac{\delta\Lambda_2}{\delta G_2^{\bullet\bullet}}\right)^{-1}\right]^{fc} G_2^{hd} \\
&\quad + \left[\left(\frac{\delta^2\Lambda_2}{\delta G_1^\bullet \delta G_1^\bullet} - 2\frac{\delta\Lambda_2}{\delta G_2^{\bullet\bullet}}\right)^{-1}\right]^{ag} G_4^{bdef} \frac{\delta^3\Lambda_2}{\delta G_2^{ef} \delta G_1^g \delta G_1^h} \left[\left(\frac{\delta^2\Lambda_2}{\delta G_1^\bullet \delta G_1^\bullet} - 2\frac{\delta\Lambda_2}{\delta G_2^{\bullet\bullet}}\right)^{-1}\right]^{hc} \\
&\quad - G_4^{abef} \frac{\delta^3\Lambda_2}{\delta G_2^{ef} \delta G_1^g \delta G_1^h} \left[\left(\frac{\delta^2\Lambda_2}{\delta G_1^\bullet \delta G_1^\bullet} - 2\frac{\delta\Lambda_2}{\delta G_2^{\bullet\bullet}}\right)^{-1}\right]^{gc} G_2^{hd} \\
&\quad - G_4^{adef} \frac{\delta^3\Lambda_2}{\delta G_2^{ef} \delta G_1^g \delta G_1^h} \left[\left(\frac{\delta^2\Lambda_2}{\delta G_1^\bullet \delta G_1^\bullet} - 2\frac{\delta\Lambda_2}{\delta G_2^{\bullet\bullet}}\right)^{-1}\right]^{gc} G_2^{hb}.
\end{aligned}
\tag{29}
$$

For an even theory also Eq. (26) simplifies:

$$
\left[\left(\frac{\delta^2\Lambda_2}{\delta G_1^\bullet \delta G_1^\bullet} - 2\frac{\delta\Lambda_2}{\delta G_2^{\bullet\bullet}}\right)^{-1}\right]^{ab} = -G_2^{ab}.
\tag{30}
$$

Combining this with Eq. (29) yields

$$
\begin{aligned}
G_4^{abcd} &= G_2^{ae} G_2^{bf} \frac{\delta^4\Lambda_2}{\delta G_1^e \delta G_1^f \delta G_1^g \delta G_1^h} G_2^{gc} G_2^{hd} + G_4^{bdef} \frac{\delta^3\Lambda_2}{\delta G_2^{ef} \delta G_1^g \delta G_1^h} G_2^{ga} G_2^{hc} \\
&\quad + G_4^{abef} \frac{\delta^3\Lambda_2}{\delta G_2^{ef} \delta G_1^g \delta G_1^h} G_2^{gc} G_2^{hd} + G_4^{adef} \frac{\delta^3\Lambda_2}{\delta G_2^{ef} \delta G_1^g \delta G_1^h} G_2^{gc} G_2^{hb}.
\end{aligned}
\tag{31}
$$

Equation (31) relates the connected four-point Green's function to derivatives of the second Legendre transform of the free energy $\Lambda_2$. In the first term, four one-point functions are *amputated* (removed by a functional derivative) from the 2PI functional $\Lambda_2$. Therefore, it retains the irreducibility property of $\Lambda_2$ and is consequently the fully 2PI four-point function that is frequently discussed in literature [22]. The three other terms are usually not discussed in literature. We replace them using the relation

$$
\frac{\delta^3\Lambda_2}{\delta G_1^a \delta G_1^b \delta G_2^{cd}} = 2\frac{\delta^2\Lambda_2^{\text{int}}}{\delta G_2^{ab} \delta G_2^{cd}},
\tag{32}
$$

which is derived in Appendix B. Since $\Lambda_2$ is the Luttinger–Ward functional of the theory, the right-hand side of Eq. (32) is simply the irreducible vertex in the $(ab)$ channel [29]. This can be inserted into Eq. (31) yielding

$$
\begin{aligned}
G_4^{abcd} &= G_2^{ae} G_2^{bf} \frac{\delta^4\Lambda_2^{\text{int}}}{\delta G_1^e \delta G_1^f \delta G_1^g \delta G_1^h} G_2^{gc} G_2^{hd} + 2\, G_4^{bdef} \frac{\delta^2\Lambda_2^{\text{int}}}{\delta G_2^{ef} \delta G_2^{gh}} G_2^{ga} G_2^{hc} \\
&\quad + 2\, G_4^{abef} \frac{\delta^2\Lambda_2^{\text{int}}}{\delta G_2^{ef} \delta G_2^{gh}} G_2^{gc} G_2^{hd} + 2\, G_4^{adef} \frac{\delta^2\Lambda_2^{\text{int}}}{\delta G_2^{ef} \delta G_2^{gh}} G_2^{gc} G_2^{hb}.
\end{aligned}
\tag{33}
$$

This finally is the parquet equation for $\Phi^4$ theory. Its graphical depiction with Feynman diagrams is shown in Fig. 6. Compared to the fermionic version of the parquet equation [7], each reducible diagram obtains an extra factor of 2 which is not present in fermionic field-theory due to the directionality of the fermionic lines. We further comment on the prefactors and perform concrete calculations of the Green's function for illustrative purposes in the Appendix C.

Figure 6: Feynman diagrams for the parquet equation [Eq. (33)]. The diagrammatic rules used in this paper are summarized in Appendix D.

The fact, that the parquet equation Eq. (33) can be obtained from Eq. (25) by performing two functional derivatives and expressing everything in terms of known quantities is the main result of this paper. We achieved our goal of relating the connected four-point Green's function $G_4$ to derivatives of the second Legendre transform of the free energy, i.e., the Luttinger–Ward functional.

## 5 Conclusion and outlook

In this work we have shown how the parquet equation [6,7] can be derived from the free-energy functional employing methods of functional analysis. This has been done for the case of $\Phi^4$ theory but can readily be extended to fermionic field theories with a two-particle scattering term by promoting the scalar field to spinors [7,29]. Even though the parquet equation is long known, its derivation so far relied on combinatorics, arguing that one can *classify* the diagrams for the full connected Green's function in terms of their two-particle reducibility without double counting or missing any [7,22,29].

The derivation shown in this work puts the parquet equation into a more general framework. In order to find relations between the full Green's function and irreducible vertex functions one can perform Legendre transformations of the free energy functional. Combining basic properties of the Legendre transformation and functional derivatives the sought after equations can be obtained. These relations can be interpreted as a recipe for adding all reducible diagrams to an irreducible vertex. In the case of the first Legendre transformation this procedure yields Dyson's equation [5]; in the case of the second transformation the parquet and Bethe–Salpeter equations are obtained. It would be interesting to continue this logic in order to classify diagrams in terms of their three-particle (ir)reducibility. The computation of three-particle diagrams has recently gained attention in the context of computing non-linear susceptibilities [24,25,34] and for accurately describing strongly correlated electrons [23]. Their classification by means of combinatorics is, however, challenging due to the large number of possibilities to draw diagrams [35].

Another advantage of the here presented derivation is that irreducible vertex functions, on which one might want to base a certain approximation, arise naturally in the theory as functional derivatives of the Legendre transformations of the free energy. For example, the fully two-particle irreducible vertex $\frac{\delta^4 \Lambda_2}{\delta G_1^a \delta G_1^b \delta G_1^c \delta G_1^d}$ that was previously introduced for combinatorial reasons [7], arises naturally as a functional derivative of the Luttinger–Ward functional. This insight might enable the derivation of equations to compute the fully two-particle irreducible vertex by performing higher-order Legendre transformations [29,36,37]. Such relations are, to the best of the authors' knowledge, currently unknown, leading to the need of employing purely numerical techniques for computing the fully two-particle irreducible vertex [14].

Furthermore, our approach makes the relation of all vertex functions to the Luttinger–Ward functional transparent potentially providing a path to deriving conserving approximations for connected Green's functions [10, 38–45]. In the case of the functional renormalization group [26, 46, 47] progress in this direction has already been made [48].

## Acknowledgments

C.E. acknowledges fruitful discussions with Alexander Herbort, Dante M. Kennes, Lennart Ronge and Michael A. Sentef. We would further like to thank Benedikt Schneider for pointing out a missing term in Eq. (25).

**Funding information** This project has been supported by the Austrian Science Funds (FWF) through projects P 32044 and V 1018.

## A  Derivation of the Bethe–Salpeter equation

In this appendix we show how to derive the Bethe–Salpeter equation (BSE) (Ref. [49], Ch. 5.3) from the inverse-second-derivative relation. We begin with an equation similar to Eq. (25):

$$\delta^{ac}\delta^{bd} + \delta^{ad}\delta^{bc} = \frac{\delta V_2^{cd}}{\delta V_2^{ab}} + \frac{\delta V_2^{dc}}{\delta V_2^{ab}}, \tag{A.1}$$

where we have symmetrized the $(cd)$ argument for later convenience. Let us look at the first term on the right-hand side

$$\frac{\delta V_2^{cd}}{\delta V_2^{ab}} = -\frac{\delta}{\delta V_2^{ab}}\frac{\delta \Lambda_2}{\delta \widetilde{G}_2^{cd}} \stackrel{\frac{\delta G_2}{\delta \widetilde{G}_2}=2}{=} -2\frac{\delta}{\delta V_2^{ab}}\frac{\delta \Lambda_2}{\delta G_2^{cd}} = -2\left(\frac{\delta G_1^e}{\delta V_2^{ab}}\frac{\delta}{\delta G_1^e} + \frac{\delta G_2^{ef}}{\delta V_2^{ab}}\frac{\delta}{\delta G_2^{ef}}\right)\frac{\delta \Lambda_2}{\delta G_2^{cd}}$$
$$= -2\frac{\delta^2 W}{\delta V_2^{ab}\,\delta V_1^e}\frac{\delta^2 \Lambda_2}{\delta G_1^e\,\delta G_2^{cd}} - 2\frac{\delta^3 W}{\delta V_2^{ab}\,\delta V_1^e\,\delta V_1^f}\frac{\delta^2 \Lambda_2}{\delta G_2^{ef}\,\delta G_2^{cd}}. \tag{A.2}$$

We neglect the first of the two resulting terms since it is one-particle reducible and will thus vanish due to symmetry (there will be no further derivatives of this term which is why we can set it to zero at this point). Furthermore, we will use

$$\frac{\delta^3 W}{\delta V_1^a\,\delta V_1^b\,\delta V_2^{cd}} = \frac{1}{2}\left(G_4^{abcd} + G_2^{ac}G_2^{bd} + G_2^{ad}G_2^{bc}\right), \tag{A.3}$$

which can easily be derived from [29]

$$\frac{\delta Z}{\delta V_n} = \frac{1}{n!}\frac{\delta^n Z}{(\delta V_1)^n}. \tag{A.4}$$

We split up $\Lambda_2$ into a non-interacting part and an interacting part according to Eq. (21). Again, note that $\Lambda_2^{\text{int}}$ is simply the Luttinger–Ward functional of the here considered $\Phi^4$ theory. Inserting this into Eq. (A.2) we get

$$\frac{\delta V_2^{cd}}{\delta V_2^{ab}} = -\left(G_4^{abef} + G_2^{ae}G_2^{bf} + G_2^{af}G_2^{be}\right)\left(-\frac{1}{2}(G_2^{-1})^{ec}(G_2^{-1})^{fd} + \frac{\delta^2 \Lambda_2^{\text{int}}}{\delta G_2^{ef}\,\delta G_2^{cd}}\right). \tag{A.5}$$

$$a \rule{}{} \blacksquare \begin{matrix} f & {-1} & d \\ & {-1} & \\ e & \end{matrix} \rule{}{} c \quad = \quad 4 \; a \begin{matrix} e & \\ \text{)i(} & d \\ f & \end{matrix} c \quad + \quad 2 \; a \blacksquare \begin{matrix} e & \\ \text{)i(} & d \\ f & \end{matrix} c$$

Figure 7: Feynman diagrams for the Bethe–Salpeter equation (A.6). The diagrammatic rules used in this paper are summarized in Appendix D.

Thus inserting this result for both terms from Eq. (A.1), reordering and using Eq. (7) we get

$$
\begin{aligned}
G_4^{abef}(G_2^{-1})^{ec}(G_2^{-1})^{fd} = {} & G_2^{ae} G_2^{bf} \frac{\delta^2 \Lambda_2^{\text{int}}}{\delta G_2^{ef}\,\delta G_2^{cd}} + G_2^{af} G_2^{be} \frac{\delta^2 \Lambda_2^{\text{int}}}{\delta G_2^{ef}\,\delta G_2^{cd}} \\
& + G_2^{ae} G_2^{bf} \frac{\delta^2 \Lambda_2^{\text{int}}}{\delta G_2^{ef}\,\delta G_2^{dc}} + G_2^{af} G_2^{be} \frac{\delta^2 \Lambda_2^{\text{int}}}{\delta G_2^{ef}\,\delta G_2^{dc}} \\
& + G_4^{abef} \frac{\delta^2 \Lambda_2^{\text{int}}}{\delta G_2^{ef}\,\delta G_2^{cd}} + G_4^{abef} \frac{\delta^2 \Lambda_2^{\text{int}}}{\delta G_2^{ef}\,\delta G_2^{dc}} \,.
\end{aligned}
\tag{A.6}
$$

As a last step we note that

$$
\frac{\delta \Lambda_2^{\text{int}}}{\delta G_2^{ab}} = \frac{\delta \Lambda_2^{\text{int}}}{\delta G_2^{ba}} \,,
\tag{A.7}
$$

which becomes apparent from Eq. (23) and the symmetry of the vertices such that we can simplify Eq. (A.6) to

$$
G_4^{abef}(G_2^{-1})^{ec}(G_2^{-1})^{fd} = 4\, G_2^{ae} G_2^{bf} \frac{\delta^2 \Lambda_2^{\text{int}}}{\delta G_2^{ef}\,\delta G_2^{cd}} + 2\, G_4^{abef} \frac{\delta^2 \Lambda_2^{\text{int}}}{\delta G_2^{ef}\,\delta G_2^{dc}} \,.
\tag{A.8}
$$

This is the familiar BSE in the case of $\Phi^4$ theory for the (ab) channel. It is graphically depicted in Fig. 7. By (ab) channel we mean the diagrams are classified into ones where the arguments (ab) and (cd) can be disconnected from each other when cutting two lines and ones where this is not possible. Note that the two-particle irreducible vertex in a specific channel here appears naturally as a second derivative of the Luttinger–Ward functional with respect to the Green's function. We would get the other channels by simply choosing different argument combinations in Eq. (A.1). As in the case of the parquet equation the above form of the BSE features extra prefactors when compared to the fermionic analogon – where these would be cancelled due to the directionality of the lines. We comment further on these prefactors and perform concrete calculations using the BSE in Appendix C.

## B Derivation of Eq. (32)

In the main part of this paper we need Eq. (32) to eliminate unknown quantities. In this appendix we show how it can be derived employing the BSE [Eq. (A.8)] from Appendix A.

We start by deriving the remaining inverse-second-derivative relations of the in total four such relations of the second Legendre transform of which we have previously written Eqs. (25) and (A.1)

$$
0 = \frac{\delta V_2^{cd}}{\delta V_1^a} \,,
\tag{B.1}
$$

$$a \underline{\phantom{xx}} \boxed{\phantom{x}} \mathbb{(} \underline{\phantom{xx}} d \\ b \underline{\phantom{xx}} \phantom{x} \underline{\phantom{xx}} c = 2 \quad a \underline{\phantom{xx}} \boxed{\mathbb{)} i \mathbb{(}} \underline{\phantom{xx}} d \\ b \underline{\phantom{xx}} \phantom{x} \underline{\phantom{xx}} c$$

Figure 8: Feynman diagrams for Eq. (B.5). The diagrammatic rules used in this paper are summarized in Appendix D.

of which we rewrite the right-hand side as

$$
\begin{aligned}
\frac{\delta V_2^{cd}}{\delta V_1^a} &= -\frac{\delta}{\delta V_1^a}\frac{\delta \Lambda_2}{\delta \widetilde{G}_2^{cd}} = -2\left(\frac{\delta G_1^e}{\delta V_1^a}\frac{\delta}{\delta G_1^e} + \frac{\delta G_2^{ef}}{\delta V_1^a}\frac{\delta}{\delta G_2^{ef}}\right)\frac{\delta \Lambda_2}{\delta G_2^{cd}} \\
&= -2G_2^{ae}\frac{\delta^2 \Lambda_2}{\delta G_1^e \delta G_2^{cd}} - 2G_3^{aef}\frac{\delta^2 \Lambda_2}{\delta G_2^{ef}\delta G_2^{cd}},
\end{aligned}
\tag{B.2}
$$

again, using $\frac{\delta G_2}{\delta \widetilde{G}_2} = 2$ which follows from the factor 2! in the definition of $\widetilde{G}_2$ in Eq. (5). We now perform a derivative with respect to $V_1$ keeping only terms that will not vanish when setting $V_1 = 0$ which gives

$$
0 = 2G_2^{ae}G_2^{bf}\frac{\delta^3 \Lambda_2}{\delta G_1^e \delta G_1^f \delta G_2^{cd}} + 2G_4^{abef}\frac{\delta^2 \Lambda_2}{\delta G_2^{ef}\delta G_2^{cd}}.
\tag{B.3}
$$

Inserting this into Eq. (B.1), canceling a factor of two and splitting up $\Lambda_2$ according to Eq. (21) we get

$$
\begin{aligned}
0 &= G_2^{ae}G_2^{bf}\frac{\delta^3 \Lambda_2}{\delta G_1^e \delta G_1^f \delta G_2^{cd}} + G_4^{abef}\left(-\frac{1}{2}(G_2^{-1})^{ec}(G_2^{-1})^{fd} + \frac{\delta^2 \Lambda_2^{\text{int}}}{\delta G_2^{ef}\delta G_2^{cd}}\right) \\
&= G_2^{ae}G_2^{bf}\frac{\delta^3 \Lambda_2}{\delta G_1^e \delta G_1^f \delta G_2^{cd}} - G_4^{abef}(G_2^{-1})^{ec}(G_2^{-1})^{fd} + G_4^{abef}\frac{\delta^2 \Lambda_2^{\text{int}}}{\delta G_2^{ef}\delta G_2^{cd}}.
\end{aligned}
\tag{B.4}
$$

We see that one can insert the BSE [Eq. (A.8)] for the last two terms which, with some argument renaming and canceling, finally yields

$$
\frac{\delta^3 \Lambda_2}{\delta G_1^a \delta G_1^b \delta G_2^{cd}} = 2\frac{\delta^2 \Lambda_2^{\text{int}}}{\delta G_2^{ab}\delta G_2^{cd}}.
\tag{B.5}
$$

This is the result used in the main part of the paper in Eq. (32). The corresponding Feynman diagrams are shown in Fig. 8.

# C  Explicit calculation of $G_4$ up to second order in the interaction

In this part we compute $G_4$ to second order in the interaction once directly performing derivatives of $W$ with respect to $V_1$ and once employing the BSE. We begin with directly differentiating $W$, computing $G_4 = \frac{\partial^4}{\partial V_1^4}W$ for which we show the relevant part of $W$ in Fig. 9.

To first order this is given by amputating the external potentials of the first diagram in Fig. 9 in all possible ways. Since there are $4! = 24$ ways to do this and the scattering potential $V_4$ is symmetric in all its arguments, the prefactor of $\frac{1}{24}$ in Fig. 9 is cancelled, and to leading order the connected two-particle Green's function is simply given by the bare interaction with external legs. This is depicted graphically in Fig. 10.

$$W = \frac{1}{24} \quad + \quad \frac{1}{16}$$

Figure 9: $W$ up to second order where we have only included diagrams with four external potentials since only these contribute to the calculation of $G_4$. Lines denote the bare Green's function.

$$\frac{\delta^4}{\delta V_1^4} \frac{1}{24} \quad = \quad {}^a\!\!\bigtimes^b_{d \quad c}$$

Figure 10: Derivative of the first order term of $W$.

For the second order we similarly perform four consecutive derivatives of the second diagram in Fig. 9. This operation is depicted diagrammatically in Fig. 11.

The prefactors can be understood as follows: The first derivative, say $\frac{\delta}{\delta V_1^a}$, produces a factor of four since it may act on any external leg. In each channel the remaining derivatives now only produce a factor of two since the two arguments on the *other side* of the argument $a$ may be exchanged. Thus, each channel is left with an overall prefactor of $\frac{1}{2}$. With this we find overall for $G_4$ up to second order in the interaction the diagrammatic result depicted in Fig. 12.

Let us turn to the calculation of $G_4$ using the BSE [Eq. (A.8)]. We first note the two diagrams contributing to the Luttinger–Ward functional up to second order that are relevant to this derivation in Fig. 13.

In order to get the leading order contribution to $G_4$, we need to perform two consecutive derivatives of the first diagram in Fig. 13. We first write this operation mathematically as

$$\frac{1}{8} \frac{\delta}{\delta G_2^{ab}} \frac{\delta}{\delta G_2^{cd}} \int \mathrm{d}(efgh)\, V_4^{efgh} G_2^{ef} G_2^{gh} = \frac{1}{8} \frac{\delta}{\delta G_2^{ab}} \left[ \int \mathrm{d}(ef)\, V_4^{efcd} G_2^{ef} + \int \mathrm{d}(gh)\, V_4^{cdgh} G_2^{gh} \right]$$
$$= \frac{1}{4} V_4^{abcd}.$$

(C.1)

In the last step we have again used the symmetry of the scattering potential $V_4$ in all its arguments. In order to obtain the correct contribution to $G_4$ from this, one still needs to attach external legs as prescribed in Eq. (A.8). Diagrammatically we denote the operations performed in Eq. (C.1) in Fig. 14.

Together with the prefactor four appearing in front of the first term on the right in Eq. (A.8) we thus obtain the correct first order to $G_4$ from the BSE. For the second order we need to perform two derivatives of the second diagram depicted in Fig. 13. Diagrammatically this operation can be denoted as done in Fig. 15

Note that here both possible orderings of arguments $(ab)$ and $(cd)$ must appear with equal weight. This becomes clear from considering Eq. (A.6) and the appearing argument symmetrization or when defining a functional derivative with respect to a function that is symmetric in its two arguments as one that is symmetrized with respect to the ordering of these two arguments. The remaining term to evaluate is the last term in Eq. (A.8). To obtain a diagram that is second order in the interaction, we connect the first order of $G_4$ as obtained previously with the first order of $\frac{\delta^2 \Lambda_2^{\mathrm{int}}}{\delta G_2^2}$. The diagrammatic result of this is shown in Fig. 16.

Together with the prefactors given in Eq. (A.8), we obtain the correct result for $G_4$ to second order in the interaction as shown in Fig. 12. In the case of the parquet equation Eq. (33) it is clear that the leading order contribution to $G_4$ originates from the derivatives of the fully

$$\frac{\delta^4}{\delta V_1^4} \frac{1}{16} \, \blacktriangleright\!\!\times\!\!\!\bigcirc\!\!\!\times\!\!\blacktriangleright \; = \; \frac{1}{2} \, {}_d^a\!\!\times\!\!\!\bigcirc\!\!\!\times\!\!{}_c^b \; + \; \frac{1}{2} \, {}_c^a\!\!\times\!\!\!\bigcirc\!\!\!\times\!\!{}_d^b \; + \; \frac{1}{2} \, {}_b^a\!\!\times\!\!\!\bigcirc\!\!\!\times\!\!{}_d^c$$

Figure 11: Derivative of the second order term of $W$.

$$G_4^{abcd} \; = \; {}_d^a\!\!\times\!\!{}_c^b \; + \; \frac{1}{2} \, {}_d^a\!\!\times\!\!\!\bigcirc\!\!\!\times\!\!{}_c^b \; + \; \frac{1}{2} \, {}_c^a\!\!\times\!\!\!\bigcirc\!\!\!\times\!\!{}_d^b \; + \; \frac{1}{2} \, {}_b^a\!\!\times\!\!\!\bigcirc\!\!\!\times\!\!{}_d^c \; + \; \mathcal{O}(V_4^3)$$

Figure 12: $G_4$ up to second order in $V_4$.

$$\Lambda_2^{\text{int}} \; = \; \frac{1}{8} \, \bigcirc\!\!\bigcirc \; + \; \frac{1}{48} \, \ominus$$

Figure 13: $\Lambda_2^{\text{int}}$ up to second order in $V_4$ only including diagrams that are relevant for the computation of $G_4$ using the BSE and the parquet equation. Note that the lines denote the full propagator in this - however replacing them by the bare one does not give corrections to $G_4$ to this order in the interaction.

two-particle irreducible vertex $\frac{\partial^4 \Lambda_2^{\text{int}}}{(\partial G_1)^4}$. The contribution is depicted in Fig. 17.

With the same combinatorial arguments as already employed in Fig. 10 and attaching external legs as prescribed by Eq. (33) we obtain the correct leading order contribution to $G_4$. The second contribution can be computed analogously to the case of the BSE using Fig. 16 with the same combinatorial factors as before. Again the additional prefactors in Eq. (33) fix the right prefactors for $G_4$ Fig. 12 while all possible argument combinations and attaching of external legs explicitly appears in Eq. (33).

$$\frac{\delta}{\delta G_2^{ab}} \frac{\delta}{\delta G_2^{cd}} \frac{1}{8} \, \bigcirc\!\!\bigcirc \; = \; \frac{1}{4} \, {}_d^a\!\!\times\!\!{}_c^b$$

Figure 14: Derivative of first-order $\Lambda_2^{\text{int}}$.

$$\frac{\delta}{\delta G_2^{ab}} \frac{\delta}{\delta G_2^{cd}} \frac{1}{48} \, \ominus \; = \; \frac{1}{8} \, {}_c^a\!\!\times\!\!\!\bigcirc\!\!\!\times\!\!{}_d^b \; + \; \frac{1}{8} \, {}_d^a\!\!\times\!\!\!\bigcirc\!\!\!\times\!\!{}_c^b$$

Figure 15: Derivative of second-order $\Lambda_2^{\text{int}}$.

$$\int G_4^{abef} \frac{\delta^2 \Lambda_2^{\text{int}}}{\delta G_2^{ef} \delta G_2^{cd}} \, \mathrm{d}(ef) \; = \; \frac{1}{4} \, {}_b^a\!\!\times\!\!\!\bigcirc\!\!\!\times\!\!{}_d^c$$

Figure 16: $G_4$ connected with $\frac{\delta^2 \Lambda_2^{\text{int}}}{\delta G_2^2}$.

$$\frac{\delta^4}{\delta G_1^4} \frac{1}{24} \, \blacklozenge\!\!\times\!\!\blacklozenge \; = \; {}_d^a\!\!\times\!\!{}_c^b$$

Figure 17: Derivative of first order $\Lambda_2^{\text{int}}$ with respect to the full 1-point function. This is the leading order to $V_4$ from the parquet equation.

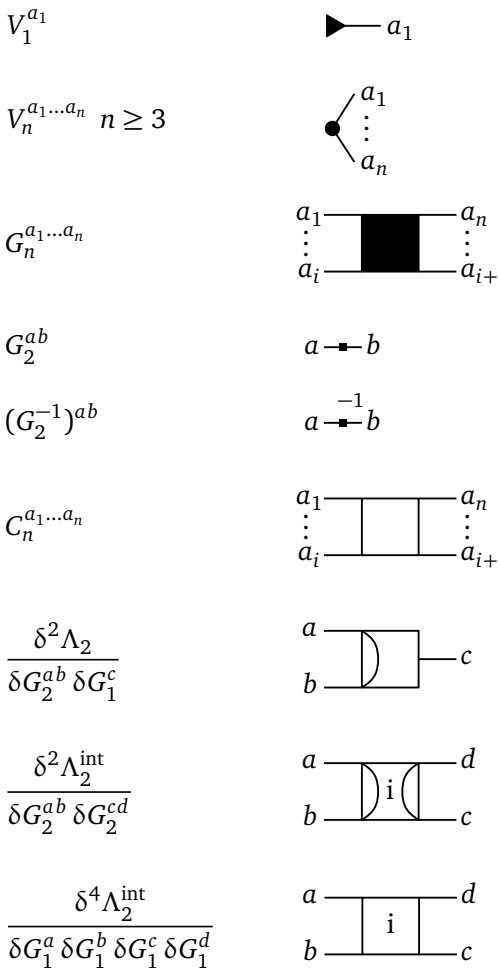

Figure 18: Diagrammatic rules for Feynman diagrams in this paper.

## D  Diagrammatic rules

Figure 18 illustrates the diagrammatic rules that are used for the Feynman diagrams in this paper. An $n$-point Green's function $G_n$ is depicted by a filled square with $n$ legs. If the square is not filled, it represents a 1PI $n$-point function $C_n$. More general derivatives of $\Lambda_2$ with respect to $G_1$ and $G_2$ are depicted with internal lines connecting the legs of the $G_2$'s. An "i" in the square means that $\Lambda^{\text{int}}$ is used in the derivative instead of the whole $\Lambda$.

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
