# Peer review of "A functional-analysis derivation of the parquet equation"

_SciPost Physics, doi:SciPost Phys. 15, 203 (2023)_

## Round 1 · Referee Report · Anonymous (Referee 1) · 2023-6-27

Report

The parquet equation is a well-known exact equation in the realm of many-body diagrammatics. The authors present an alternative derivation of this equation using a functional integral approach. In Secs. 2 and 3, starting from a functional integral representation of the partition function, they express many-body Green functions as (functional) derivatives wrt source fields. They recapitulate that the Legendre transform wrt these source fields yields a generating functional for one-particle irreducible vertex functions, and they re-rederive relations such as Eq. (13). These results are well-known in the literature.

In Sec. 4, they generalize their approach and set up a generating functional for two-particle irreducible vertex functions. This is still well-known. They then use generalizations of, e.g., Eq. (13) to re-derive the Parquet equation.

The novel derivation of the Parquet is to the best of my knowledge sound, and I did find it elegant. This is an interesting result. I was a little sceptical whether this result alone (re-derivation of a known equation using different means) indeed warrants publication in SciPost because I am not sure what one can really learn from this (the functional integral itself formalism is well-known).

All things considered, I do recommend publication.
  • validity: -
  • significance: -
  • originality: -
  • clarity: -
  • formatting: -
  • grammar: -

Author:  Christian J. Eckhardt  on 2023-09-15  [id 3981]

(in reply to Report 1 on 2023-06-27)

We thank the referee for the precise recapitulation of our work and their recommendation for publication in SciPost physics.We would like to comment on the question, what one can learn from our rederivation of the parquet equation.

First of all, as referee 3 has pointed out, a derivation based on the path-integral and mathematical methodology can be viewed as somewhat conceptually deeper than the previously employed combinatorial arguments and may therefore have intrinsic value.

Furthermore, our derivation places the parquet equation into a broader context which might enable the derivation of new relations. For example, our derivation yields the fully 2-particle irreducible vertex as a derivative of the Luttinger-Ward functional. This might enable the derivation of new diagrammatic relations for this crucial quantity.

Another important point is that the scheme we have put forward in our work is of general nature and may be adapted to derive further equations for higher order vertex functions. In such cases, combinatorial arguments are likely to become too complicated to employ effectively, necessitating a method based on calculations.

As a last point, our derivation may be of value to newcomers to the field to whom the many combinatorial arguments, that are sometimes hard to verify for oneself, can seem overwhelming.

Overall we hope that these points illustrate the benefits of our derivation and the overall value of our work.

---

## Round 1 · Referee Report · Anonymous (Referee 2) · 2023-6-29

Strengths

Elegant derivation of an important self-consistency equation of many body physics.

Weaknesses

Relation to previous literature (work by Vasiliev) would be nice to be clarified or discussed.

Report

The manuscript by Eckhart et al. describes a derivation of the parquet equations from the second functional Legendre transform of the free energy. The manuscript is very clearly written and the result is worth being published, as it provides a systematic derivation of this important self-consistency equation. I have only minor comments that I hope are useful for the authors and support the publication of this interesting manuscript.

Minor points:

  1. In Eq. (32) the authors should recheck the indices e in the last two terms. Should they be c?

Also why do they write the same term twice on the left hand side? Should maybe one of them be differentiated by \delta G^{dc} (instead of \delta G^{cd})? Maybe adding a comment in the text would help.

  1. Conceptually I wonder why the authors derive the parquet equations from the second order Legendre transform instead of the full transform (forth transform): For simplicity consider an even theory with V_1 = V_3 = 0. Then the full transform would be

\Lambda(~G_2, ~G_4) = W - V_2 ~G_2 - V_4 ~G_4

with stationarity conditions for V_2 and V_4. Also compare Martin & De Dominicis 1964.

The reason why I am asking is because the Legendre transform can be considered the tool to define an ensemble with fixed statistics to a certain moment; the full transform fixing the second and the fourth moment of the theory. As such one expects to obtain a self-consistency equation where the right hand side depends on the second and forth moment given by the left hand side.

Also following the book by Vasiliev, Ref 29 of the manuscript, p. 251, these authors state that the parquet equations follow from the full transform by iterating their Eq. 6.113. I would be very much interested in the authors view on this point. Does their approach and the one by Vasiliev yield the very same parquet equation?

  • validity: top
  • significance: high
  • originality: high
  • clarity: top
  • formatting: perfect
  • grammar: perfect

Author:  Christian J. Eckhardt  on 2023-08-11  [id 3900]

(in reply to Report 2 on 2023-06-29)
Category:
answer to question

We sincerely thank the referee for their positive report and recommendation for publication of our manuscript in SciPost physics after minor revisions.

1.) Indeed Eq.(32) contained typos, it will be replaced in the next version.

2.) This is a very interesting and intricate point. Our understanding of this is the following: In order to derive the parquet equation we focus on the (ir)reducibility properties of diagrams. The parquet equation classifies diagrams in terms of 2-particle (ir)reducibility and therefore the second order Legendre-transformation seems the appropriate one to achieve such a classification.

One can nicely illustrate this on the example of the first transform: In Eq.(16) we have repeated the well known derivation of Dyson's equation. This is also achieved via the first order transform, even though it relates the full 2-point Green's function to the 2-point vertex. Interestingly, the first order transform, however, does not give a 'good recipe' for computing the full 2-point vertex. Performing a double derivative in G1 is cumbersome since one usually does not write down diagrammatic expansions including G1. Here the second order Legendre-transformation offers a solution through Eq.(23) that is identical to the Ward-identity (insert Eq.(21) into Eq.(23) to see this identity). Here the self-energy is much more conveniently expressed using a derivative with respect to the full Green's function.

For the case of 2-particle (ir)reducibility, we see the parquet-equation (and also the BSE to that matter) as the 2-particle analogon to Dyson's equation (just as a comment: for the BSE this is a well established notion while for the parquet equation this notion seems to be somewhat new). While it achieves to relate different vertex functions in a closed form, there is no good recipe to compute the fully-irreducible 2-particle vertex since it is again given in terms of derivatives with respect to 1-point functions.
One may now speculate that in order to get such a 'good recipe' the next Legendre-transform will be needed: namely that with respect to V4. In this case one may evaluate the analogous Legendre-property to Eq.(23) and hope to get an analogon to the Ward-Identity for the full 2-particle irreducible vertex. This is exactly what Eq.6.113 (and also Eq.6.112) in the book of Vasiliev do - even though the concrete relation to the fully-irreducible 2-particle vertex does not become clear from their equations. However, Vasiliev does *not* derive the parquet equation in the sense that a closed relation between full vertex functions is never obtained. The classification of diagrams is still only done on a combinatorial level (also see text below Eq.6.113 of his book).

We hope that these explanations clarify the relation of our work to that of Vasiliev and are happy to answer further questions on this matter.
In the manuscript we tried to make the notion that the parquet equation and the BSE are the analogon to Dyson's equation on the 2-particle level clear in the discussion section. Beyond this we feel that the above discussion might be too involved to be added to the paper in reasonable form and on top might contain too much speculation. But we are open for suggestions on how to include it.

---

## Round 1 · Referee Report · Anonymous (Referee 3) · 2023-7-9

Report

The authors present a functional-analysis derivation of the parquet equation. The parquet equation is used in many instances of condensed-matter field theory. It is routinely motivated by diagrammatic arguments. A purely algebraic or functional — and thus arguably conceptually deeper — derivation has been a longstanding problem. The present work constitutes a breakthrough in that regard, will likely become a standard reference in the field, and is therefore suitable for publication in SciPost Physics.

The presentation of the manuscript is good but can be improved, in order for the paper to not only be a reference that is cited because of its title, but to convey important insight to practitioners of as well as newcomers to the parquet formalism. Suggestions for improvement are elaborated below.

Major points:

  • The target readership for this paper are condensed-matter field theorists using the parquet formalism for interacting electrons. For this reason, the authors ultimately set V_1 = V_3 = 0. However, this information appears very late in the manuscript and should rather be explained from the outset. Moreover, while V_1 (and V_2) are needed for the Legendre transformations, what purpose does V_3 serve, if it is never used in the transformations and set to zero at the end? This should also be explained in the beginning. The authors might want to reconsider if it is better to keep V_3 for completeness or rather remove it to have more compact equations and diagrammatic expansions (without V_3, the number of terms in Figs. 1 and 2 shrinks drastically). Obviously, one can imagine compromises, such as moving the equations with V_3 to an appendix or supplement while reducing the main text to the relevant part without V_3.

  • Condensed-matter field theorists using the parquet equations for interacting electrons less often use diagrammatic expansions for the free energy or Legendre transformations thereof, but rather for the self-energy and the two-particle vertex. Therefore, I strongly suggest to add to each diagrammatic expansion of a generating functional the corresponding diagrammatic expansion for the expectation values of interest. In that way, Fig. 1 could also contain the expansions for \tilde{G}_2, \tilde{G}_4, G_2, and G_4; Fig. 2 the expansions for C_2 and C_4. Importantly, there should also be a diagrammatic expansion of \Lambda_2^{int} (Luttinger-Ward functional). Added to that could be the expansion of the self-energy as the derivative o the Luttinger-Ward functional w.r.t. the full propagator, etc.

  • It is stated that Eq. (33) is obtained by adding Eq. (31) to itself with exchanged arguments. Shouldn’t it rather be by averaging Eq. (31) and itself with exchanged arguments? Thereby, shouldn’t there be factors of 1/2 in all terms except the one on the LHS and the first on the RHS?

  • This leads to the important point of factors of 1/2. It is rather strange that the full (disconnected and connected) propagator and the connected propagator have different prefactors. I believe that \tilde{G}_2 = < \Phi^2 >, reducing to -1/V_2 at V_1=V_3=V_4=0, is more conventional than \tilde{G}_2 = 1/2 < \Phi^2 >. More importantly, I am unsure whether the factors of 1/2 are correct in the final equations. Typically, the Bethe-Salpeter equations for a real phi^4 theory each have a prefactor 1/2. Furthermore, it seems unusual to me that, even in real phi^4 theory, the parquet equation has two terms per channel. Can one not use the symmetry of the vertices to reduce the number of terms to one per channel? I encourage the authors to carefully check the issue of factors of 1/2 and to give references to the literature on, e.g., the Bethe-Salpeter equations in real phi^4 theory and other established equations.

Minor points:

  • above Eq. (1): Phi^3 + Phi^4 theory?

  • Eq. (1): Einstein’s summation convention (regarding a_i) is already used here and not only from Eq. (9) (where it is mentioned) onward.

  • Below Eq. (10): connected -> 1PI.

  • Fig. 8 lacks a definition of V_2. An explicit definition of G_1 would also be helpful.

  • validity: -
  • significance: -
  • originality: -
  • clarity: -
  • formatting: -
  • grammar: -

Author:  Christian J. Eckhardt  on 2023-09-15  [id 3983]

(in reply to Report 3 on 2023-07-09)

We thank the referee for the overall positive assessment of our work, writing that it 'constitutes a break-through' and 'will likely become a standard reference' in the field.
The referee has, however, raised several important points to improve the presentation of the results. We will answer to these points in detail below and have updated the manuscript to incorporate them.

-We agree with the referee that readers coming from fermionic field theory might not find the inclusion of V_3 insightful. We have therefore removed it from the manuscript - as suggested by the referee - now limiting ourselves to Phi^4 theory.

-Indeed we had not included the diagrams for the Luttinger-Ward functional in the previous version of the paper which we have now done. Concerning the expansion of other diagrammatic quantities, we have added an extra appendix showing a concrete calculation to obtain the diagrams for G_4, once using direct functional derivatives of W and once using the BSE or the parquet equation. This gives the concrete diagrammatic form for the quantities appearing in the BSE and the parquet equation up to second order in the interaction V_4. We hope that this helps to clarify the concrete diagrammatic form of several 2-particle functions.
We have, however, not included expansions for further 1-particle functions since the paper is mainly concerned with the derivation of 2-particle quantities.

-Eq.(33) in the previous version of the paper indeed contained a typo which does not appear in the revised version - we thank the referee for pointing this out

-We thank the referee for the suggestion to simplify the final equation using symmetry and explaining all occuring prefactors. We have updated the final equations also taking the symmetry of the vertex functions into account which now yields one diagram per channel in both the parquet equation and the BSE. Instead, a prefactor now appears for the correct counting of diagrams.
Concerning factors of 1/2: First of all it is true that we have defined Tilde{G} with an extra prefactor when compared to the disconnected Green's functions usually reported in literature. This is however important in order for Tilde{G} to serve as a variable in the Legendre transformations. We have commented on this in the main text below Eq.(5) of the revised manuscript.
Concerning factors of 1/2 in the final equations we have now included an extra appendix where we perform concrete calculations of G_4 up to second order in the interaction which illustrates how one obtains the correct prefactors of the diagrams in G_4 including the prefactors appearing in the BSE and the parquet equation.
The reason these appear somewhat different from standard literature might have to do with the definition of the functional derivative with respect to the full Green's function. When removing a Green's function 'diagrammatically' from a diagram one may remove it in both possible directions. This corresponds to a definition of the functional derivative that comprises the sum of the two derivatives with both arguments exchanged. Mathematically, this definition is somewhat unintuitive which is why we have not adopted it in the calculation. A specific example where this can be seen is Eq.(47) in the new appendix while Fig.(14) shows the diagrammatic analogue to the operation.

-We now consider Phi^4 theory only which we state above Eq.(1)
-Indeed we use Einstein's summation convention already in Eq.(1) which we now explicitly state
-We corrected the typo below Eq.(10)
-The definition of G_1 is implicit via the definition of G_n. We do not include the definition of the bare Green's function since we have used a single line somewhat ambiguous since we also use it to signify external arguments of vertex functions. We thus explicitly state the usage of undressed Green's functions in the captions of the figures where they appear.

Overall we think that the comments of the referee have significantly improved the paper for which we would like to thank them. We hope that the updated presentation of the results is found to be more straightforward.

---

## Round 2 · Referee Report · Anonymous (Referee 1) · 2023-9-25

Report

The revised version can be published as it is.

---

## Round 2 · Referee Report · Anonymous (Referee 3) · 2023-10-29

Report

The revised version of the manuscript has improved in terms of presentation. I will leave a few more suggestions for improvement. After the authors have considered these suggestions, the paper can be published from my point of view.

  • I repeat my suggestion to give at least one precise (with equation number) reference to a published version of the BSEs of real phi^4 theory.
  • Fig. 18 contains a symbol for V_n, n \geq 3. This can be simplified since only V_4 appears in the new version.
  • The meaning of the diagrams is not entirely clear due to the minimalistic character of App. D and the choice of the authors to use "somewhat ambiguous" notation in the diagrams. Why not draw amputated legs shorter than attached legs? Why not use different colors / linestyles / etc. for bare vs. full propagators? To exemplify my confusion: (i) I suppose the first term on the RHS of Fig. 1 is 1/2 V_1^a G_0^{ab} V_1^b. What is the first term on the RHS of Fig. 2? 1/2 G_1^a G_2^{ab} G_1^b can't be correct? (ii) By inserting lowest-order vertices in Fig. 6, one should obtain Fig. 12. How do the prefactors 2 become prefactors 1/2? It seems that redefining a new channel-dependent 2PI vertex equal to 1/4 of the previous channel-dependent 2PI vertex might be helpful (affecting, e.g., the last three terms on the RHS of Fig. 6 and the two terms on the RHS of Fig. 7)?
  • The authors occasionally write that, e.g., G_3=0 due to V_3=0. But this also requires V_1=0, doesn't it?

Typos: - paragraph below Eq. (33): appendix Appendix -> Appendix - Eq. (47) LHS: G_2^{ef} G_2^{ef} -> G_2^{ef} G_2^{gh} - Eq. (47) RHS, 2nd line: G_4^{abcd} -> V_4^{abcd} - paragraph below Fig. 15: "as one that this" -> "as one that is"

---

## Round 2 · List of Changes

• Removed V_3 from the original action and all diagrams subsequently.
  • Added explaining remark for prefactor in Tilde{G}
  • Used symmetry of vertices originating from the symmetry of the 1-particle Green's function under exchange of its arguments. This leads to a simplification of the parquet equation and the Bethe-Salpeter equation and their corresponding diagrams. Extra prefactors now appear.
  • Added an extra appendix for the calculation of the 2-particle Green's function by different means. This in particular illustrates the origin of the prefactors in the parquet equation and the Bethe-Salpeter equation as well as giving concrete hands on examples on how to perform calculations.
  • The other appendices have also been updated to employ the symmetry of vertices more effectively.

---

## Editorial Decision

published